# Reproducibility Report:
# Contextualizing Hate Speech Classifiers with Post-hoc Explanation

## Reproducibility Summary

*The presented report evaluates Contextualizing Hate Speech Classifiers with Post-hoc Explanation Kennedy et al. (2020) paper within the scope of ML Reproducibility Challenge 2020. Our work focuses on both aspects constituting the paper: the method itself and the validity of the stated results. In the following sections, we have described the paper, related works, algorithmic frameworks, our experiments and evaluations.*

**Scope of Reproducibility**

For the GHC (a dataset), the most important difference between BERT+WR and BERT+SOC is the increase in recall. While, for Stormfront (a dataset), there are similar improvements for in-domain data and the NYT dataset. But, for verifying the claims we also have tried to run the same experiment on a new data-set.

**Methodology**

We have tried to re-implement the author's code and verify the claims made in their original paper. We have experimented on NVIDIA Tesla GPU which was less efficient than the original author's resource (NVIDIA GeForceRTX 2080 Ti).

**Results**

We have able to reproduce claims as mentioned in the following section 2 (Scope of Reproducibility) marked as point 2 and 3. But we are not on the same page with the authors for a few reported experiments mentioned as point 1 and 4 in the same section.

**What was easy**

The original authors provide code for most of the experiments presented in the paper. The code was easy to run and allowed us to verify the correctness of our re-implementation. The explanations in the code made the work pretty easy for us.

**What was difficult**

Training of the models was very time taking as we had to wait for hours to train the model and the resources used by the original authors are not readily available everywhere.

**Communication with original authors**

We were in contact with the second author via E-mail, as he was responsive and shared details that were not explicitly mentioned in the paper.

# 1   Introduction

Hate-speech classification comes under larger efforts to reduce the damage caused by offensive and oppressive language Waldron (2012); Gelber and McNamara (2016). While the relative sparsity of hate speech necessitates sampling using keywords Olteanu et al. (2018) or a selection from environments with very high rates of hate-speech de Gibert et al. (2018), the performance has increased with access to more sophisticated algorithms and data. Mondal et al. (2017); Silva et al. (2016). Thus, present-day text classifiers struggles with learning a model of hateful speech that generalizes to applications in real-world Wiegand et al. (2019). The over-sensitivity of neural hate speech classifiers to group identifiers like "Jews," "black," and "gay," classifies to hate speech when used in the correct context, is a particular issue. Dixon et al. (2018). The performance of neural text classifiers in detecting hateful speech is state-of-the-art, but they are uninterpretable and could break if given an unexpected input data. Niven and Kao (2019). Hence not easy to contextualize the method of the model to identifying words. To estimate model agnostic and context-independent post-hoc feature importance, the author uses explanation algorithm of Sampling and Occlusion (SOC). Jin et al. (2020). They used the SOC explanation algorithm on the Gab Hate Dataset Kennedy et al. (2020), a new data-set for "hate-based rhetoric", and the Stormfront dataset which is the largest white nationalists online community, characterised by pseudo-rational discussions on race de Gibert et al. (2018). Using the SOC information, which revealed that models are biased with respect to group identifiers, therefore they suggested a new approach based on regularization to improve the model's sensitivity towards the group identifiers surrounded by context. They regulate the group identifiers importance during training, forcing models for investigation of the context in which they operate. They discovered that regularisation reduces the importance of group identifiers while increasing the importance of hate speech's more generalizable features, such as dehumanising and abusive language. They found that regularisation significantly decreases the false positive rate in studies on an out-of-domain news article's test-set comprising group identifiers that are heuristically expected as "non-hate" speech. Concurrently, out-of-sample classification performance for in-domain is either maintained or enhanced.

# 2   Scope of reproducibility

The paper here points out that most of the Hate Speech classifiers available now are majorly tilted or over-sensitive to some of the identifiers or words like (gay, black, and Muslim) but they don't take into account the fact that the mere presence of the word would not make it oppressive but the context in which it is used gives us the correct classification. If the context is not taken into account then many samples would result in false positives. Thereby, reducing the accuracy. The work here is formulated to detect hate speech as disambiguating the use of offensive words from abusive versus non-abusive contexts. We plan to use the code that is available from the authors themselves and then as per the paper we will be reviewing and testing the claims made. Some of the major claims of the paper are:

1. In GHC dataset, the most significant difference between BERT+WR and BERT+SOC is the increase in recall.

2. For Stormfront (a dataset), same improvements is seen for in-domain data and the NYT dataset.

3. Paper claims performance for their proposed method as (Precision = 56.11, Recall = 54.23, F1 = 54.71 and NYT Acc = 93.89) on average

4. The efficiency claimed in the paper is as follows (BERT = 5:1 mins, BERT+OC = 13:36 mins, BERT+SOC = 19:3 mins)

We have tried to verify the above claims made in the paper using the data-sets presented by the original authors and as well as on a new data-set. To train the model the authors have used GeForce RTX 2080 Ti GPU, which we tried to implement using our institutional resources.

# 3   Methodology

The authors in their previous paper Jin et al. (2020) have explained methods which are used in the current paper. We here first explain the parts of the previous paper and then show how it is used in the current paper. The methods and approach is described below:

## 3.1   Model descriptions

### 3.1.1   Context-Independent Importance (CII)

Given a phrase p := $x_{i:j}$ appearing in a specific input $x_{1:T}$, first the setting is relaxed and then they define the importance of a phrase independent of contexts of length N adjacent to it. For an intuitive example, to evaluate the CII up to one

word of very in the sentence The film is very interesting in a sentiment analysis model, then sample some possible adjacent words, and average the prediction difference after some practice of masking the word very (as shown in Figure 1 below). The N-context independent importance is formally written in Equation 1.

$$\phi(p, \hat{x}) = E_{x_\delta}[s(x_{-\delta}; \hat{x}_\delta) - s(x_{-\delta} \backslash p; \hat{x}_\delta)] \tag{1}$$

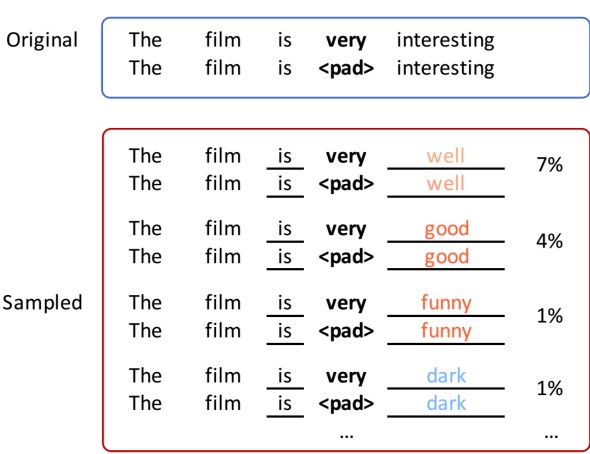

Figure 1: Word Masking and Value Prediction

where $x_{-\delta}$ denotes the resulting sequence after masking out a context of length N surrounding the phrase p from the input x. Here, $\hat{x}_\delta$ is a sequence of length N sampled from a distribution $p(\hat{x}_\delta | x_{-\delta})$, which is conditioned on the phrase p as well as other words in the sentence x. Accordingly, they use $s(x_{-\delta}; \hat{x}_\delta)$ to denote the model prediction score after replacing the masked-out context $x_{-\delta}$ with a sampled context $\hat{x}_\delta$. $x \backslash p$ is used to denote the operation of masking out the phrase p from the input sentence x. Following the notion of N-CII, they define CII of a phrase p by increasing the size of the context N to sufficiently large ( e.g., length of the sentence). The CII can be equivalently written as given in Equation 2.

$$\phi^g(p) = E_x[s(x) - s(x \backslash p) | p \subseteq x] \tag{2}$$

| muslim jew jews white islam blacks muslims women whites gay black democat islamic allah jewish lesbian transgender race brown woman mexican religion homosexual homosexuality africans |
| --- |

Figure 2: 25 group identifiers selected from top weighted words in the TF-IDF BOW linear classifier on the GHC

### 3.1.2 Model Interpretation

To assess the issue in depth, they explore hate speech models' bias towards group identifiers and why that leads to false-positive errors during prediction. Then they examine the models themselves to see how sensitive models are to group identifiers. Linear classifiers can be examined in terms of their most highly-weighted features. Then, for the task of extracting comparable information from the fine-tuned methods discussed above, a post-hoc explanation algorithm is used. They gathered a set of twenty-five identity words from the top features in a bag-of-words logistic regression of hate speech $GHC_{train}$, which they use in subsequent analyses.

**Explanation-based measures:** BERT models can model complex word and phrase compositions; for example, some words are only offensive when used with particular ethnic groups. Sampling and Occlusion (SOC) algorithm is used to capture this, which is capable of generating hierarchical explanations for a prediction. SOC begins by assigning importance scores to sentences in such a manner that compositional effects between the phrase and the context $x_\delta$ around it are eliminated. SOC assigns an importance score $\phi(p)$ where p is a phrase in a sentence x to show how the phrase contributes to the sentence being classified as hate speech. Then, in the 2-way classifier, the algorithm computes the difference of the unnormalized prediction score $s(x)$ between "hate" and "non-hate." The algorithm then calculates

99 the average change in $s(x)$ for different inputs when the phrase is masked with padding tokens (noted as $x\backslash p$), in which
100 the N-word contexts around the phrase $p$ are sampled from a pre-trained language model, while other words remain the
101 same as the given $x$. Formally, the importance score $\phi(p)$ is measured as given in Equation 3.

$$\phi(p) = E_{x_\delta}[s(x) - s(x\backslash p)] \tag{3}$$

102 Meanwhile, the SOC algorithm generates a hierarchical layout by performing agglomerative clustering over explanations.
103 Then, they compute average word importance using SOC explanations from $GHCtest$ and present the top 20 in Figure.
104 4.

---

jew jews mexican blacks jewish brown black mus-
lim homosexual islam

---

Figure 3: 10 group identifiers selected for the Stormfront dataset

**Bias in Prediction:** Models of hate speech can be overly sensitive to group identifiers. They create an adversarial test
106 set of New York Times (NYT) articles that are filtered to contain a balanced, random sample of the twenty-five (GHC
107 Dataset) and ten (Stromfront dataset) group identifiers, as shown in Figure 2 and Figure 3 respectively, to provide an
108 external measure of models' over-sensitivity to group identifiers.

| **BERT** | $\Delta$ **Rank** | **Reg.** | $\Delta$ **Rank** |
|---|---|---|---|
| ni**er | +0 | ni**er | +0 |
| ni**ers | -7 | fag | +35 |
| kike | -90 | traitor | +38 |
| mosques | -260 | faggot | +5 |
| ni**a | -269 | bastard | +814 |
| jews | -773 | blamed | +294 |
| kikes | -190 | alive | +1013 |
| nihon | -515 | prostitute | +56 |
| faggot | +5 | ni**ers | -7 |
| nip | -314 | undermine | +442 |
| islam | -882 | punished | +491 |
| homosexuality | -1368 | infection | +2556 |
| nuke | -129 | accusing | +2408 |
| niro | -734 | jaggot | +8 |
| muhammad | -635 | poisoned | +357 |
| faggots | -128 | shitskin | +62 |
| nitrous | -597 | ought | +229 |
| mexican | -51 | rotting | +358 |
| negro | -346 | stayed | +5606 |
| muslim | -1855 | destroys | +1448 |

Figure 4: Top 20 words by mean SOC weight before (BERT) and after (Reg.) regularization for GHC

109 Models must not ignore identifiers, but rather match them to the appropriate context. Figure 5 illustrates the effect of
110 ignoring identifiers by removing random subsets of words ranging in size from 0 to 25, with each subset sample size
111 repeated five times. On the NYT dataset, lower rates of false positives are accompanied by poor hate speech detection
112 performance.

**Explanation Regularization:** Given that SOC explanations are differentiable fully, at the time of training, the SOC
114 explanations on the group identifiers are regularized to be close to 0 in addition to the classification objective $\mathcal{L}'$. The
115 combined learning objective is by the following Equation 4.

$$\mathcal{L} = \mathcal{L}' + \alpha \sum_{w \in x \cap S} [\phi(w)]^2 \tag{4}$$

116 where $S$ denotes the set of group names and $x$ denotes the word sequence to be input. The strength of the regularisation
117 is determined by the hyper-parameter $\alpha$. They also experiment with regularising input occlusion (OC) explanations,
118 which is specified as the change in prediction when a word or phrase is masked out, avoiding the sampling step in SOC.

119 **Visualizing Effects of Regularization:** The effect of regularization can be seen by considering Figure 5. Here
120 visualization of SOC hierarchically clustered explanations before and after regularization are done to correct the false
121 positive predictions.

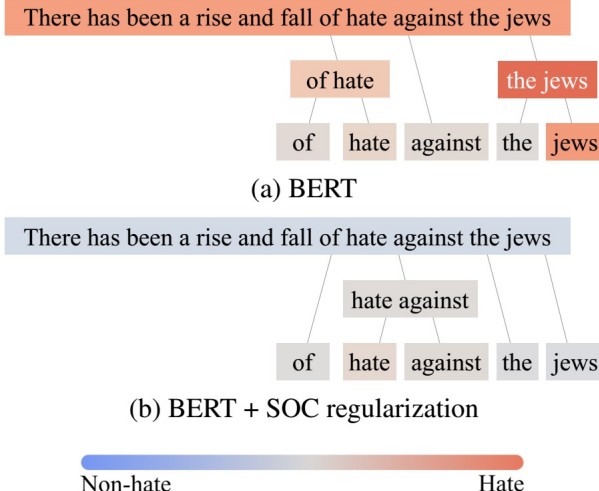

(a) BERT

(b) BERT + SOC regularization

Non-hate            Hate

Figure 5: Hierarchical explanations of test example of GHC dataset before and after explanation regularization to
correct the false positive predictions

## 3.2 Datasets

123 The original authors chose two publicly available dataset for the experiments that features the logical parts of hate-
124 speech, versus only the use of explicitly hostile language and slurs. The "Gab Hate Corpus" Kennedy et al. (2020)
125 is a huge dataset with arbitrary 27,655 example, which have been annotated on as per the typology of "hate based
126 manner of speaking", motivated by the criminal codes of hate-speech outside the U.S. also, research of sociology on
127 bias and dehumanization. A social network Gab contains high pace of "hate discourse" Zannettou et al. (2018); Lima
128 et al. (2018) and populated by the "Extreme right" Anthony (2016); Benson (2016). Likewise with deference to area
129 and definitions de Gibert et al. (2018) annotated and sampled posts of "Stormfront" web space Meddaugh and Kay
130 (2009) and annotated at the label of sentence as per a comparable annotation guide as utilized in the GHC dataset.

Table 1: GHC Dataset

| GHC | Total | Hate | Non Hate |
|---|---|---|---|
| Train | 24,353 | 2,027 | 22,326 |
| Test | 1,586 | 372 | 1,214 |

Table 2: Stromfront and New (Twitter hate-speech) Dataset

| | Stromfront Dataset | | | New Twitter hate-speech Data | | |
|---|---|---|---|---|---|---|
| | Total | Hate | Non-Hate | Total | Hate | Non-Hate |
| Train | 7,896 | 1,059 | 6,837 | 6,555 | 780 | 5,775 |
| Test | 1,998 | 246 | 1,752 | 1,634 | 196 | 1,438 |
| Validation | 979 | 122 | 857 | 1,156 | 140 | 1,016 |

131 Train set and test set were randomly produced by the authors for the Stormfront dataset (80/20), as mentioned in their
132 paper with "hate" as a +ve label, and the test set was made by the authors from the GHC dataset by picking random
133 stratified data regarding the "target population" tag (potential qualities including race/identity target, sexual religious and
134 so forth). A solitary "hate" mark was made by picking the association of the 2 fundamental labels, "human degradation"
135 and "calls for violence". Training set of the GHC contains 24,353 posts with 2,027 marked as "Hate", and test set of

the GHC contains 1,586 posts with 372 marked as "Hate". Out of 7,896 posts in the training set of Stormfront dataset, 1,059 marked as hate, out of 979 posts, 122 marked as hate in the validation set, and out of 1,998 posts, 246 marked as hate in the test dataset. We have trained the model on our new Twitter hate-speech dataset taken from Kaggle [1]. Train set of new Twitter hate-speech dataset (new train) contains 6,555 posts with 780 marked as "Hate", test set for the (new test) contains 1,634 posts with 196 marked as "Hate", and validation set for the (new val) contains 1,156 posts with 140 marked as "Hate". Table 1 presents the number of "hate" and "non hate" labels of GHC Dataset. Table 2 shows the number of "hate" and "non hate" labels in Stormfront dataset as well as in Twitter hate-speech dataset. We have made the new Twitter hate-speech dataset in such a way that it contains similar percent of "hate" and "non hate" labels compared to Stormfront dataset. The Figure 6 shows the comparison of old vs new dataset.

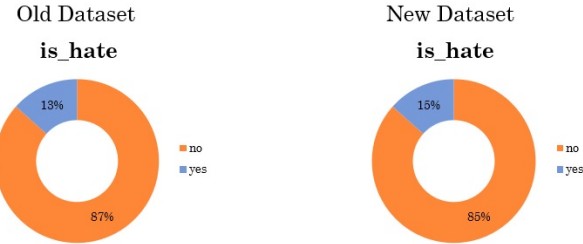

Figure 6: Old(Stormfront) vs New(Twitter) Dataset Comparison

## 3.3  Computational requirements

The authors have used GPU GeForce RTX 2080 Ti for training the model. The training times for the authors for BERT+OC and BERT+SOC were only 2 times and 4 times respectively greater than that of the BERT. Whereas we have experimented on NVIDIA Tesla GPU. The detailed comparisons of GPU and time are shown in Table 3 and 4. The authors framework were far superior to ours which may be the reason that their training time and usage are more efficient than ours.

Table 3: GPU Comparisons

| GPU Features | Paper Report | Our Framework |
|---|---|---|
| GPU Name | TU102 | GK110B |
| GPU Details | NVIDIA GeForce RTX 2080 Ti | NVIDIA Tesla K40m |
| Memory Size | 11 GB | 12 GB |
| Memory Clock | 14 Gbps | 6 Gbps |
| Memory Type | GDDR6 | GDDR5 |

Table 4: Time Comparisons

| Methods | Approach | Training Time (per epoch) | GPU memory use |
|---|---|---|---|
| BERT | Paper | 5 m 1 s | 9095M |
| | Ours | 15 m 13 s | 7253M |
| BERT + OC | Paper | 12 m 36 s | 9411M |
| | Ours | 21 m 5 s | 7041M |
| BERT + SOC | Paper | 19 m 3 s | 9725M |
| | Ours | 24 m 33 s | 7352M |

## 4  Reimplementation of code

This section shortly summarizes the main structure of the code accompanying this reproducibility check. The authors' code was largely used as the starting point for our reimplementation in PyTorch and various other python libraries (like

---

[1]https://www.kaggle.com/vkrahul/twitter-hate-speech

numpy, scikit-learn, scikit-image, matplotlib and torchtext). We fine-tuned over the BERTbase model using the public code [2].

# 5 Results

We have investigated different methods such as BERT, Word identifiers removal before BERT training (BERT+WR), BERT with regularizing occlusion (BERT+OC) and BERT with regularizing sampling and occlusion (BERT+SOC) with similar parameter and hyper-parameter values as described by the authors. We have also used the NYT test set as blind dataset to measure how good a model has learnt the contexts with the group identifiers for hate speech. Experiment has been done on the GHC, Stormfront and external labelled Twitter hate-speech dataset for evaluating the classification of hate speech in-domain and accuracy on the test set of NYT. We have used the same 25 terms (for GHC); 10 terms for Stormfront as in the paper. Accordingly, for the Stormfront dataset we have filtered the NYT dataset to have these 10 terms (N = 5,000).

Table 5: F1-score, Recall, Precision and their respective standard deviations on test set of Stormfront and accuracy on evaluation set of NYT

| **Stormfront Dataset** | | | | | |
|---|---|---|---|---|---|
| **Method** | **Approach** | **Precision** | **Recall** | **F1-Score** | **NYT-Accuracy** |
| BERT | Paper | $57.76 \pm 3.9$ | $54.43 \pm 8.1$ | $55.44 \pm 2.9$ | $92.29 \pm 4.1$ |
| | Ours | $55.81 \pm 2.3$ | $57.68 \pm 5.7$ | $56.54 \pm 1.7$ | $91.87 \pm 2.6$ |
| BERT+WR | Paper | $53.16 \pm 4.3$ | $57.16 \pm 5.7$ | $54.60 \pm 1.7$ | $92.47 \pm 3.4$ |
| | Ours | $55.76 \pm 3.1$ | $56.21 \pm 7.2$ | $55.87 \pm 1.5$ | $93.53 \pm 3.2$ |
| BERT + OC ($\alpha = 0.1$) | Paper | $57.47 \pm 3.7$ | $51.10 \pm 4.4$ | $53.82 \pm 1.3$ | $95.39 \pm 2.3$ |
| | Ours | $56.74 \pm 3.2$ | $53.44 \pm 6.1$ | $55.24 \pm 3.4$ | $92.56 \pm 4.7$ |
| BERT + SOC ($\alpha = 1.0$) | Paper | $56.05 \pm 3.7$ | $54.35 \pm 3.4$ | $54.97 \pm 1.1$ | $95.40 \pm 2.0$ |
| | Ours | $61.87 \pm 5.8$ | $51.78 \pm 1.1$ | $56.93 \pm 4.5$ | $90.86 \pm 2.8$ |

Performances (as reported in this paper and what we obtained during reproducibility experiment) are shown in Table 5 and Table 6 for Stromfront and GHC dataset respectively. We have reported standard deviation and mean for the performances for 10 executions of BERT+SOC (as reported in paper), BERT + OC, BERT + WR and BERT. We have tested the reproduced results also. Though our reproduced results are comparable as per reported in the paper for most of the methods in Stromfront datasets but we obtain lower precision, recall and F1-score for GHC dataset (BERT+SOC with $\alpha = 0.1$). Testing on blind dataset NYT is comparable for most of the cases. Only in few cases our reproduced results differ from the paper's reported range values for Stormfront dataset like higher precision (+ 9%) and lower accuracy (- 5%) in BERT+SOC ($\alpha = 1.0$).

Table 6: F1-score, Recall, Precision and their respective standard deviations on test set of GHC and accuracy on evaluation set of NYT

| **GHC Dataset** | | | | | |
|---|---|---|---|---|---|
| **Method** | **Approach** | **Precision** | **Recall** | **F1-Score** | **NYT-Accuracy** |
| BERT | Paper | $69.87 \pm 1.7$ | $66.83 \pm 7.0$ | $67.91 \pm 3.1$ | $77.79 \pm 4.8$ |
| | Ours | $64.91 \pm 2.8$ | $57.67 \pm 6.7$ | $60.14 \pm 7.1$ | $70.48 \pm 4.7$ |
| BERT+WR | Paper | $67.61 \pm 2.8$ | $60.08 \pm 6.6$ | $63.44 \pm 3.1$ | $89.78 \pm 3.8$ |
| | Ours | $59.76 \pm 8.1$ | $55.98 \pm 4.3$ | $57.84 \pm 3.6$ | $84.35 \pm 3.2$ |
| BERT + OC ($\alpha = 0.1$) | Paper | $60.56 \pm 1.8$ | $69.72 \pm 3.6$ | $64.14 \pm 3.2$ | $89.43 \pm 4.3$ |
| | Ours | $49.78 \pm 9.5$ | $60.34 \pm 6.3$ | $56.45 \pm 7.2$ | $90.23 \pm 1.1$ |
| BERT + SOC ($\alpha = 0.1$) | Paper | $70.17 \pm 2.5$ | $69.03 \pm 3.0$ | $69.52 \pm 1.3$ | $83.16 \pm 5.0$ |
| | Ours | $62.48 \pm 5.2$ | $66.21 \pm 6.5$ | $64.24 \pm 3.4$ | $74.56 \pm 5.7$ |

---

[2]https://github.com/owaisCS/TestHateSpeech

Table 7: Precision, Recall, F1-Score (%) on New Twitter test set

| Data Set | Metrics | BERT + SOC ($\alpha = 1.0$) | BERT + OC ($\alpha = 0.1$) | BERT + WR | BERT |
|---|---|---|---|---|---|
| | | Ours | Ours | Ours | Ours |
| **Twitter Hate-speech** | Precision | $80.61 \pm 3.9$ | $84.74 \pm 5.8$ | $50.71 \pm 3.9$ | $49.36 \pm 2.3$ |
| | Recall | $56.42 \pm 3.4$ | $58.32 \pm 4.3$ | $54.68 \pm 5.6$ | $52.75 \pm 5.7$ |
| | F1-Score | $66.38 \pm 1.1$ | $69.09 \pm 2.6$ | $49.35 \pm 1.9$ | $51.58 \pm 1.3$ |

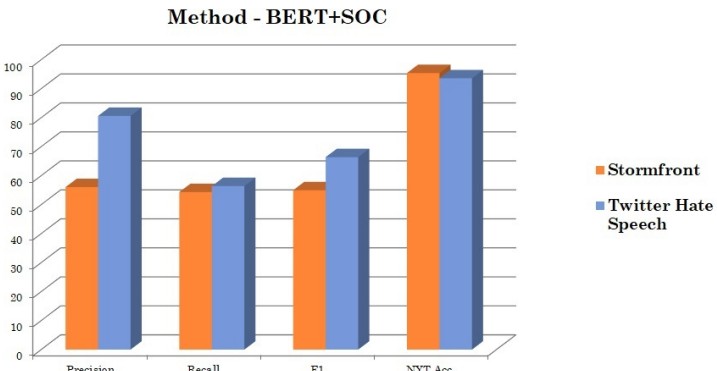

Figure 7: Comparison of Metrics for BERT+SOC ($\alpha$=1.0) between Stromfront and Twitter hate-speech Dataset

Table 7 shows precision, recall and f1-score obtained by BERT+SOC ($\alpha = 1.0$), BERT+OC ($\alpha = 0.1$), BERT +WR and BERT methods on new Twitter hate corpus. Comparisons of different metrics on new Twitter hate-speech dataset and Stromfront dataset are shown in Figure 7 which shows significant increase of precision ( 20%) on new Twitter hate-speech dataset compared to Stromfront using BERT+SOC ($\alpha$=1.0).

## 6   Discussion

We have able to reproduce few claims as reported by the authors in the paper - (i) For Stormfront (a dataset), same improvements are seen for in-domain data as well as NYT and (ii) The authors claims some performances such as (Precision = 56.11, Recall = 54.23, F1 = 54.71 and NYT Acc = 93.89) on average. But we are not in the same page with the authors for a few reported experiments - In GHC dataset, the main difference between BERT+SOC and BERT+WR is the increase in recall as we have obtained lower precision, recall and f1-accuracy. This may be due the experimental framework differences. Due to a bar on time, we could not run the BERT+SOC several times to make the comparison more detailed. In the future, we would also try to verify their claims using similar GPU configurations and incorporate more new datasets.

### 6.1   What was easy

The authors' code which was publicly available, covered almost all the experiments in their paper. It also helped us to validate the correctness of our replicated codebase. The link to our code is stated in section 4 and additionally, the original paper is quite complete, straightforward to follow, and the ReadMe file in their project helped a lot.

### 6.2   What was difficult

For replicating the experiments one will need the GPU similar to the one used by the original authors or it will be difficult to get results on time as was in our case.

### 6.3   Communication with original authors

While working on the challenge, we stood in E-mail contact with the second author and want to thank the author for his responsive communication, which helped us to clarify a great deal of implementation and evaluation specifics. For example, which particular BERT model from the library was used by them to train the model. We also got the data-sets that they used to carry out the experiments. The communication with the author helped us a lot in understanding the paper.

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
