# OpenReview forum: "Reproducibility Report: Contextualizing Hate Speech Classifiers with Post-hoc Explanation"
_ML_Reproducibility_Challenge/2020 — Reject_

### Official Review · AnonReviewer1 · 2021-02-28
**Reproducibility Report Contextualizing Hate Speech Classifiers with Post-hoc Explanation**

**Rating:** 8
**Confidence:** 4

**Review:**

The authors attempted to reproduce the results from the paper "Contextualizing Hate Speech Classifiers with Post-hoc Explanation" by Kennedy et al. 2020.

+ The authors attempted to reproduce several claims from the original paper.
+ they address the fact that their computational set up was different and in fact slower than that reported in the original paper
+The authors provide an explanation about the models in Section 3 before describing their reproducibility experiments.

- It was not clear if or how the authors reproduced the train/val/test sets from the original paper. In one case, they state that "Train and test parts were arbitrarily produced for Stormfront sentences" and similarly for the other datasets.
- In addition, while the original paper reports performance measures along with confidence intervals (e.g. 57.76 ± 3.9 for precision), the current paper reports the performance measure. So for the claims that are stated as not reproducible  it was not clear whether the performance measures were within the confidence intervals.

- minor issues: in several places the "instruction text" seems to mistakenly remain. e.g. lines 145 -147. "Provide information on computational requirements for each of your experiments. For example, the number of
147 CPU/GPU hours and memory requirements. You’ll need to think about this ahead of time, and write your code in a
148 way that captures this information so you can later add it to this section. "
- some minor typos in the paper that should be corrected prior to acceptance. e.g. "Sromfront" line 166.




**Familiar With The Original Paper:**

I have read the original paper

**Reproducibility Summary:**

Report has summary

---

### Official Review · AnonReviewer3 · 2021-03-01
**Method and result validity evaluation of Kennedy et al (2020) through reproducing the study.**

**Rating:** 4
**Confidence:** 4

**Review:**

The authors evaluate both the method and validity of the results in their reproduction of Kennedy et al (2020). Results of  some of the experiments could be replicated. But some others did not yield comparable results with the original paper.

The authors run the method on a new dataset as well. However, it is not clear how a baseline method would perform on this new dataset.

The code is not provided! “The code was easy to run and  allowed us to verify the correctness of our re-implementation.” But all of your experiments on the code provided by Kennedy et al (2020) (this one stated in the paper as well). If you are not using or providing your code, why do you say this? How can we verify this point?

There are many copy paste from the original paper, e.g. “We chose two public corpora for our experiments which feature the logical parts of hate speech, versus only the use of  slurs and explicitly hostile language …”.  This should be paraphrased or just summarized much better. Changing some words with their synonyms is neither paraphrasing nor summarization. In the example sentence, the word “we” causes confusion about what you and original authors do. The other example in this line is the aforementioned “implementing the code”

Number formatting, e.g., 7896 -> 7,896

Why do you not have the standard deviation for your results in Table 6? It is not clear what is happening in the paragraph you explain Tables 6 and 7.

“have able to reproduced” -> have able to reproduce


**Familiar With The Original Paper:**

I have read the original paper

**Reproducibility Summary:**

Report has summary

---

### Decision · Program_Chairs · 2021-03-31

**Decision:**

Reject

**Comment:**

Overall reviews and/or the paper content not good enough for the AC to recommend to the journal.